# GPT-Critic: Offline Reinforcement Learning for End-to-End Task-Oriented Dialogue Systems

**Youngsoo Jang[1], Jongmin Lee[1], Kee-Eung Kim[1,2]**
[1]School of Computing, KAIST, Daejeon, Republic of Korea
[2]Graduate School of AI, KAIST, Daejeon, Republic of Korea
{ysjang,jmlee}@ai.kaist.ac.kr, kekim@kaist.ac.kr

## Abstract

Training a task-oriented dialogue agent can be naturally formulated as offline reinforcement learning (RL) problem, where the agent aims to learn a conversational strategy to achieve user goals, only from a dialogue corpus. It is very challenging in terms of RL since the natural language action space is astronomical, while feasible (syntactically and semantically correct) actions are very sparse. Thus, standard RL methods easily fail and generate responses diverging from human language, even when fine-tuning a powerful pre-trained language model. In this paper, we introduce GPT-Critic, an offline RL method for task-oriented dialogue. GPT-Critic is built upon GPT-2, fine-tuning the language model through behavior cloning of the critic-guided self-generated sentences. GPT-Critic is essentially free from the issue of diverging from human language since it learns from the sentences sampled from the pre-trained language model. In the experiments, we demonstrate that our algorithm outperforms the state-of-the-art in the task-oriented dialogue benchmarks including MultiWOZ 2.0 and ConvLab.

## 1 Introduction

Building an end-to-end task-oriented dialogue agent is one of the promising applications of natural language processing (NLP) tasks, yet challenging due to large language action spaces and limited availability of human-annotated data. Recently, large-scale pre-trained language models (LM) have achieved remarkable successes in various NLP tasks with prohibitively large vocabulary (Devlin et al., 2019; Radford et al., 2019; Brown et al., 2020; Raffel et al., 2019). The current best performing end-to-end conversational agents for a task-oriented dialogue system utilize a pre-training on large-scale corpus and fine-tuning on downstream tasks (Ham et al., 2020; Yang et al., 2021; Lin et al., 2020; Peng et al., 2021). This combination of pre-training and fine-tuning significantly improves overall performance in the task-oriented dialogues. However, supervised fine-tuning (i.e. imitation learning of the dialogue corpus) alone may not be sufficient to learn an optimal dialogue strategy since the corpus often contains suboptimal dialogues collected from human participants of diverse expertise levels. Thus, in order to optimize the task performance of the conversational agent, goal-oriented training (i.e. reinforcement learning) is an essential and promising direction to pursue.

Training a task-oriented conversational agent from a dialogue corpus can be naturally formulated as offline reinforcement learning (RL) problem (Levine et al., 2020; Fujimoto et al., 2019; Jaques et al., 2020), which offers the prospect to optimize the policy solely from the fixed dataset without online environment interaction. Most of the existing offline RL methods are built on the off-policy Actor-Critic framework, which performs iterative optimization of the policy (i.e. actor) and the action-value function (i.e. critic) (Fujimoto et al., 2019; Janner et al., 2019; Kumar et al., 2020). Yet, a naive application of these offline RL methods generally results in poor dialogue strategies which generate responses in no way similar to human language (Lewis et al., 2017; Zhao et al., 2019; Jang et al., 2020).

Weighted behavior cloning (BC) (Wang et al., 2020) is one of the representative offline RL algorithms, which is free from the issue of diverging from human language. Weighted BC amounts

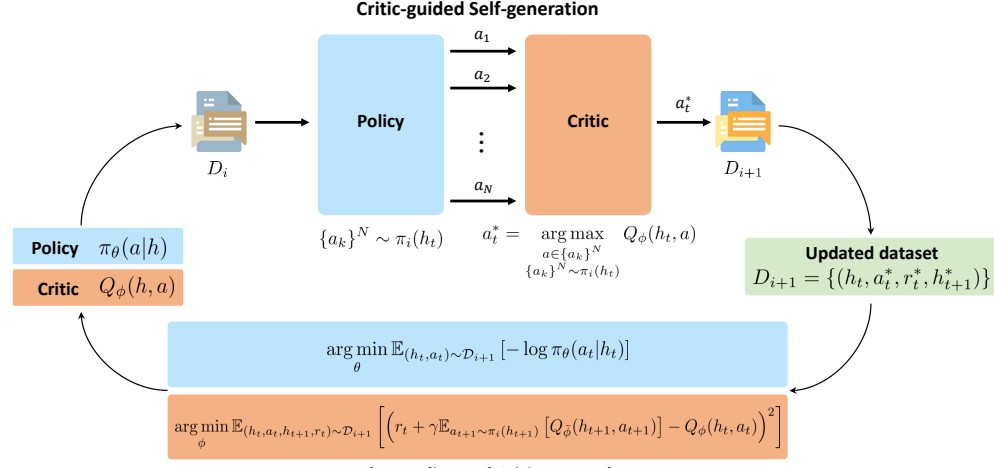

Figure 1: Overall architecture of GPT-Critic.

to filtering out bad actions and imitating good actions. In the context of task-oriented dialogues, that would be equivalent to simply dropping the unsuccessful dialogues from the corpus. However, dropping a whole dialogue from training would be wasteful, since they may still contain some task-specific information that is useful to properly respond to user requests in the intermediate steps.

In this paper, we present an offline RL algorithm for task-oriented dialogue, which can be adopted for any generative pre-trained language model. Our algorithm, GPT-Critic, aims to revise unsuccessful dialogues into successful ones, rather than removing them as done in weighted BC. It starts with fine-tuning the GPT-2 model and learning the action-value function (critic) using the dialogue corpus. Then, GPT-Critic generates a strategically promising action that is selected based on the value estimated by the critic. GPT-Critic updates the policy through behavior cloning of the critic-guided self-generated responses. This is in contrast to the previous methods that perform weighted behavior cloning on the dialogue corpus, where the action choice is restricted to the support in the dataset (Wang et al., 2020). Compared to traditional actor-critic methods, since GPT-Critic does not rely on policy gradient and updates the policy within the support of generated actions from the GPT-2, it thus inherits GPT-2's ability to generate human-like responses. In the experiments, we demonstrate that GPT-Critic outperforms the state-of-the-art end-to-end dialogue agent in the task-oriented dialogue benchmarks including MultiWOZ 2.0 (Budzianowski et al., 2018) and ConvLab (Zhu et al., 2020).

## 2 BACKGROUND

### 2.1 OFFLINE REINFORCEMENT LEARNING FOR TASK-ORIENTED DIALOGUES

We consider the task-oriented dialogue system that can be modeled as a partially observable Markov decision process (POMDP) (Williams & Young, 2007) defined by tuple $\langle S, A, O, T, Z, R, \gamma \rangle$ where $\mathcal{S}$ is the set of environment states $s = \langle g, h \rangle$ (underlying state that consists of the user goal $g$ and dialogue history $h$), $A$ is the set of actions $a$ (a sequence of tokens which represents `dialogue act` and `system response`), $O$ is the set of observations $o$ (user utterance), $T(s'|s, a) = \Pr(s_{t+1} = s'|s_t = s, a_t = a)$ is the transition function, $Z(o|s', a) = \Pr(o_{t+1} = o|s_{t+1} = s', a_t = a)$ is the observation probability, $R(g, h, a)$ is the reward function indicating the utility of executing action $a$ in history $h$ and the user goal $g$, and $\gamma \in (0, 1)$ is a discount factor. The history at time step $t$, $h_t = \{o_0, a_0, \ldots o_{t-1}, a_{t-1}, o_t\}$, is a sequence of all previous observations and actions. Since the underlying state $s$ (e.g. user goal) is not directly observable, the agent makes decisions based on the entire observation-action history. The policy $\pi(a_t|h_t)$ is mapping from history $h_t$ to a probability distribution over $A$. The goal is to find an optimal policy $\pi^*$ that maximizes the expected cumulative rewards, i.e. $\pi^* = \arg\max_\pi \mathbb{E}_\pi \left[ \sum_{t=0}^\infty \gamma^t R(g, h_t, a_t) \right]$. The action-value function of policy $\pi$ is defined as $Q^\pi(h, a) := \mathbb{E}_\pi \left[ \sum_{t=0}^\infty \gamma^t R(g, h_t, a_t)|h_0 = h, a_0 = a \right]$, where $Q^\pi$ is a unique solution of the Bellman equation: $Q^\pi(h, a) = \mathbb{E}_g[R(g, h, a)] + \gamma \mathbb{E}_\pi [Q^\pi(h', a')]$.

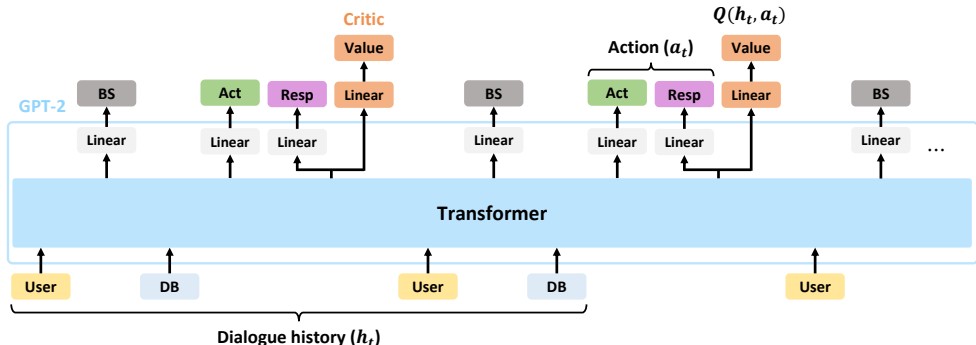

Figure 2: Architecture of policy and critic network based on GPT-2.

Using offline RL for dialogue policy optimization, the agent optimizes the policy from the pre-collected dataset $\mathcal{D} = \{\{(g^j, h_t^j, a_t^j, r_t^j, h_{t+1}^j)_{t=0}^T\}_{j=1}^N\}$ without online environment interaction during the intermediate stages of training. Prior offline RL algorithms (Fujimoto et al., 2019; Janner et al., 2019; Kumar et al., 2020) rely on *off-policy* actor-critic method, where the critic network is trained by minimizing the temporal differnce error with respect to the target policy $\pi$:

$$\arg\min_\phi \mathbb{E}_{(h_t, a_t, r_t, h_{t+1}) \sim \mathcal{D}} \left[ \left( r_t + \gamma \mathbb{E}_{a_{t+1} \sim \pi(h_{t+1})} \left[ Q_{\bar\phi}(h_{t+1}, a_{t+1}) \right] - Q_\phi(h_t, a_t) \right)^2 \right] \quad (1)$$

where $\bar\phi$ is the parameters of the target network. As discussed in the prior work (Fujimoto et al., 2019; Kumar et al., 2020), optimizing this loss can be challenging in the offline RL setting due to the overestimation issue in the bootstrapping process by taking out-of-distribution (OOD) actions to evaluate the value of the next state.

## 2.2 END-TO-END TASK-ORIENTED DIALOGUE SYSTEM

We focus on the MultiWOZ 2.0 dataset (Budzianowski et al., 2018), which is a representative benchmark for task-oriented dialogue. The MultiWOZ dataset is a fully-annotated corpus of human-human task-oriented conversations, which is collected via the Wizard-of-Oz setting (Kelley, 1984). The traditional approach to building a task-oriented dialogue system adopts a modular pipeline, which consists of the following four modules: 1) A natural language understanding (NLU) module (Kim et al., 2017; Zhu et al., 2020) identifies the user's intent and extracts the information of slots and their values, 2) A Dialogue state tracking (DST) module (Williams et al., 2013) infers the belief state, 3) A dialogue policy (POL) module decides the system action, 4) A natural language generation (NLG) module (Wen et al., 2015) generates the system response corresponding to the system action. Recently, end-to-end task-oriented dialogue methods leveraging the pre-trained language model have been proposed (Yang et al., 2021; Ham et al., 2020; Lin et al., 2020; Peng et al., 2021; Hosseini-Asl et al., 2020), and significantly improves overall performance in the task-oriented dialogues. In this paper, our algorithm is built upon UBAR (Yang et al., 2021), which is based on GPT-2 (Radford et al., 2019) and currently the state-of-the-art end-to-end dialogue agent for the MultiWOZ domain.

## 3 OFFLINE REINFORCEMENT LEARNING FOR END-TO-END TASK-ORIENTED DIALOGUE SYSTEMS

The corpus collected from human-human conversations inevitably contains unsuccessful dialogues in terms of task completion. For example, approximately 20% dialogues of the MultiWOZ dataset fail to meet the user goal. Therefore, a naive behavior cloning of the whole dataset would limit the performance of the conversational agent since the dataset includes a lot of unsuccessful dialogues: an agent that imitates failure would be inevitably suboptimal. Yet, dropping the unsuccessful dialogues from the corpus as done in weighted BC is also undesirable, since they may contain some task-specific information that is useful to properly respond to user requests. We thus aim to *revise* unsuccessful dialogues into successful ones in order to prevent repeating the past failure while improving the task performance.

In this section, we present GPT-Critic, an offline RL algorithm for task-oriented dialogue. Our GPT-Critic is analogous to Actor-Critic method: GPT (Actor) decides which action to take while the Critic informs how good the action was and provides a signal for policy improvement. Still, GPT-Critic is distinct from the Actor-Critic methods in that it does not rely on the policy gradients, which are generally known to cause the issue of diverging from human language (Lewis et al., 2017; Zhao et al., 2019). Instead, we sample a set of action candidates using GPT-2 and pick the best one using the critic, which constitutes a *revised* dialogue corpus. Then, we perform supervised fine-tuning of the GPT-2 on the revised dialogue corpus. This learning procedure of our GPT-Critic does not hurt the agent's capability to generate human-like sentences, given that the generated action candidates were all natural-looking sentences due to the power of large pre-trained LM. Our algorithm is built upon the GPT-2 but it can be adopted for any generative pre-trained language model.

## 3.1 POLICY EVALUATION

Our GPT-Critic starts by training the action-value function (i.e. critic), which can evaluate the candidates for the response. The architecture of the critic network basically follows GPT-2 with employing different last layers to compute the Q-value. The parameterization of the critic network $Q_\phi$ is designed to share the parameters of the Transformer (Vaswani et al., 2017) layers of GPT-2, where the parameters of the Transformer layers are only updated during the policy improvement step. The critic network is trained by minimizing the temporal difference error with respect to the dataset $\mathcal{D}$:

$$\arg\min_\phi \mathbb{E}_{(h_t, a_t, r_t, h_{t+1}, a_{t+1}) \sim \mathcal{D}} \left[ \left( r_t + \gamma Q_{\bar{\phi}}(h_{t+1}, a_{t+1}) - Q_\phi(h_t, a_t) \right)^2 \right] \tag{2}$$

where $\bar{\phi}$ is the parameters of the target network. Note that Eq. (2) is an *on-policy* evaluation on the dataset $D$, which can be optimized very stably since every $a_{t+1}$ is always an *in-distribution* sample of $D$. This is in contrast to Eq. (1), which requires evaluation of *out-of-distribution* actions sampled from the target policy $\pi$. The OOD action-value estimation can be very unreliable if the target policy deviates much from the dataset.

This kind of on-policy evaluation has been explored in the offline RL context for stable policy optimization (Brandfonbrener et al., 2021; Goo & Niekum, 2021), but they are limited to only *one-step* policy improvement: once the policy $\pi$ is improved by the initial on-policy $Q$-function (i.e. $\pi(s) = \arg\max_a Q(s, a)$), the new policy deviates from the dataset policy, thus it requires off-policy evaluation for further policy iteration. In contrast, our GPT-Critic performs policy improvement by *generating an improved dataset* based on the learned critic, where we can perform on-policy evaluation on the new dataset again. As a consequence, GPT-Critic can enjoy the stable *multi-step* policy iteration through alternation between on-policy evaluation and policy improvement via revising dataset, which will be discussed in the following section.

## 3.2 POLICY IMPROVEMENT VIA DATASET REVISION

In the task-oriented dialogues, the reward is given by the external program provided as a part of the dataset, which checks whether the user goal is satisfied by examining the dialogue history. To generate the improved dataset, we adopt the common automatic evaluation of dialogue systems, where the agent generates dialogue act and system response on every system turn with fixed user utterances. More formally, the GPT-Critic generates a new dataset containing *revised* responses by:

$$\mathcal{D}_{i+1} = \{(g, h_t, a_t^*, r_t^*, h_{t+1}^*) \mid a_t^* = \underset{\substack{a \in \{a_k\}^N \\ \{a_k\}^N \sim \pi_\theta^i(h_t)}}{\arg\max} \ Q_\phi(h_t, a) \text{ where } h_t \in \mathcal{D}_i\} \tag{3}$$

where $\{a_k\}^N$ is a set of $N$ response candidates generated from the policy $\pi$ (i.e. fine-tuned GPT-2), and $\mathcal{D}_i$ is the dataset at $i$-th iteration. In the task-oriented dialogues, a reward function $R(g, h, a)$ is provided that can compute a reward given a user goal, dialogue history, and system action. The revised reward $r_t^* = R(g, h_t, a_t^*)$ is computed by given user goal, dialogue history, and revised system action $a_t^*$. The dialogue history is a sequence of all previous observations and actions, thus the revised history $h_{t+1}^* = \{o_0, a_0, \dots, o_t, a_t^*, o_{t+1}\}$ is defined by replacing the original action $a_t$ of $h_{t+1}$ with the revised action $a_t^*$. The examples of revised responses can be found in Appendix B.

In order to address the prohibitively large language action spaces, we explicitly consider the set of response candidates that are generated from the fine-tuned GPT-2. The GPT-Critic selects the

---

**Algorithm 1** GPT-Critic

---

**Input:** Training dataset $\mathcal{D}_0 = \{\{(g^j, h_t^j, a_t^j, r_t^j, h_{t+1}^j)_{t=0}^T\}_{j=1}^N\}$, policy network (GPT) $\pi_\theta$, critic network $Q_\phi$
Fine-tune the initial policy represented by GPT-2 model (e.g. UBAR)
**for** each iteration $i$ **do**

Update critic by minimizing the temporal difference error until convergence:

$$\arg\min_\phi \mathbb{E}_{(g,h_t,a_t,r_t,h_{t+1},a_{t+1})\sim\mathcal{D}_i} \left[ \left( r_t + \gamma Q_{\bar{\phi}}(h_{t+1}, a_{t+1}) - Q_\phi(h_t, a_t) \right)^2 \right]$$

Update dataset by critic-guided self-generation:

$$\mathcal{D}_{i+1} = \{(g, h_t, a_t^*, r_t^*, h_{t+1}^*) \mid a_t^* = \arg\max_{\substack{a \in \{a_i\}^N \\ \{a_i\}^N \sim \pi_\theta^i(h_t)}} Q_\phi(h_t, a) \text{ where } h_t \in \mathcal{D}_i\}$$

Update policy by behavior cloning of critic-guided self-generated dataset:
(Early stop according to the loss on the validation set)

$$\arg\min_\theta \mathbb{E}_{(h_t,a_t)\sim\mathcal{D}_{i+1}} \left[ -\log \pi_\theta(a_t|h_t) \right]$$

**end for**

---

most promising response by calculating the Q-values over the response candidates. GPT-Critic then performs behavior cloning of critic-guided self-generated dialogues:

$$\arg\min_\theta \mathbb{E}_{(h_t,a_t)\sim\mathcal{D}_{i+1}} \left[ -\log \pi_\theta(a_t|h_t) \right] \tag{4}$$

where $\theta$ is the parameters of GPT-2. The policy improvement of GPT-Critic is performed by behavior cloning of generated dialogues from the GPT-2, thus GPT-Critic inherits GPT-2's ability to generate human-like responses.

We can theoretically show that the updated policy by the above policy improvement step has a higher value than the old policy. Furthermore, we can also theoretically show that updated policy by the higher number of candidate actions has a higher value than the policy updated by the lower number of candidate actions. We formalize this result in Theorem 1.

**Theorem 1.** *(Policy Improvement) Given a policy $\pi$ and the number of sampling actions $N \geq 1$, If we update the new policy $\pi_N^{\text{new}}$ by*

$$\forall s, \pi_N^{\text{new}}(\cdot|s) = \arg\max_{\substack{a \in \{a_k\}^N \\ \{a_k\}^N \sim \pi(s)}} Q^\pi(s, a)$$

*then $Q^{\pi_N^{\text{new}}}(s, a) \geq Q^\pi(s, a) \; \forall s, a$ always holds. Furthermore, for any $N, M$ such that $N \geq M \geq 1$, $Q^{\pi_N^{\text{new}}}(s, a) \geq Q^{\pi_M^{\text{new}}}(s, a) \; \forall s, a$ always holds. (Proof in Appendix A.)*

We describe our algorithm, GPT-Critic, in Algorithm 1, that alternates between policy evaluation and policy improvement via revising the dataset until the policy performance converges.

## 4 RELATED WORK

**End-to-End Task-Oriented Dialogue Systems.** The traditional approach to building a task-oriented dialogue system adopts a modular pipeline, which consists of natural language understanding, dialogue state tracking, dialogue policy, and natural language generation. Recently, pre-trained LM-based end-to-end task-oriented dialogue agents that all sub-tasks recast as a single sequence prediction problem have been proposed (Ham et al., 2020; Hosseini-Asl et al., 2020), and significantly improved overall performance in the task-oriented dialogues. There are a number of variants of GPT-2-based end-to-end task-oriented dialogue agents. Yang et al. (2021) leverage the entire dialogue session of every dialogue turn. Peng et al. (2021) adopt transfer learning and machine teaching for training a GPT-2-based dialogue agent. Lin et al. (2020) present efficient dialogue state tracking with a minimal generation length, then leverage pre-trained language models for task-oriented dialogues.

**Reinforcement Learning for Task-Oriented Dialogue Systems.** Applying the standard RL methods straightforwardly to optimize a task-oriented dialogue agent causes the issue of diverging from human language. To address this problem, interleaving reinforcement learning with supervised learning has been proposed but it is still not free from the issue of diverging from human language (Lewis et al., 2017). Recently, the latent representation models for language actions have been introduced to address the aforementioned problem (Zhao et al., 2019; Yarats & Lewis, 2018). They disentangle the semantics of the utterance and the natural language generation, and then perform goal-based training in the space of the latent variables instead of directly optimizing utterances. However, they cannot be directly applied to large-scale pre-trained language models that are not designed in a way that works inherently with discrete latent variables. Jaques et al. (2020) use KL-control to restrict the policy to stay close to its prior policy, but it still suffers from divergence from human language even with carefully chosen hyper-parameters. Furthermore, Jang et al. (2020) adopt Bayes-adaptive Monte-Carlo planning to negotiation dialogue then use it as a policy improvement operator. This approach can prevent the issue of diverging from human language through the policy improvement based on behavior cloning of self-generated dialogues. However, they assume a user model that is difficult enough to be considered another problem.

**Offline Reinforcement Learning.** There have been extensive studies on offline RL (Fujimoto et al., 2019; Levine et al., 2020; Kumar et al., 2020; Wang et al., 2020). Most of prior works are built on the off-policy actor-critic framework, and they focus on the overestimation issue by taking the OOD actions (Kumar et al., 2019; Lee et al., 2020; Fujimoto et al., 2019; Jaques et al., 2020; Kumar et al., 2020). However, a naive application of these offline RL methods suffer from the issue of diverging from human language in the task-oriented dialogues (Lewis et al., 2017; Zhao et al., 2019; Jang et al., 2020). On the other hand, there are a number of recent works on weighted behavior cloning, where a policy is trained by a variant of supervised learning loss (Wang et al., 2020; Peng et al., 2019; Siegel et al., 2020). The weighted behavior cloning approaches filter out bad actions, then perform behavior cloning on high-quality data. However, in the task-oriented dialogues, simply dropping the unsuccessful dialogues from the corpus is undesirable, since they may contain some task-specific information that is useful to properly respond to user requests. Our GPT-Critic aims to *revise* unsuccessful dialogues into successful ones, which is in contrast to the weighted behavior cloning on the fixed training dataset, where the action choice is restricted to the support in the dataset (Wang et al., 2020; Peng et al., 2019; Siegel et al., 2020). More recently, Chen et al. (2021) introduce Decision Transformer, a Transformer-based architecture that casts the problem of RL as conditional sequence modeling. These offline RL methods based on behavior cloning are directly applied to the task-oriented dialogues without aforementioned issue, but their results are similar to that of behavior cloning in the task-oriented dialogues.

## 5 EXPERIMENTS

In this section, we show the experimental results of GPT-critic on both automatic evaluation and human evaluation. First, we evaluate the performances of GPT-Critic on the MultiWOZ 2.0 (Budzianowski et al., 2018) as dataset-based automatic evaluation, compared with baseline methods including offline RL algorithms. Second, for more realistic evaluation, we conduct a simulator-based evaluation on the ConvLab framework (Zhu et al., 2020). Third, we also conduct the human evaluation to evaluate the quality of generated responses. Finally, we give a qualitative analysis of our method using generated dialogue examples on the training dataset of MultiWOZ 2.0, which shows how GPT-Critic improves the performance through the behavior cloning of self-generated dialogues. The qualitative analysis with generated dialogue examples can be found in Appendix B.

### 5.1 EXPERIMENTAL SETUP

We implement GPT-Critic based on the HuggingFace Transformers library (Wolf et al., 2019) and codebase of UBAR (Yang et al., 2021), which is a GPT-2-based current state-of-the-art end-to-end task-oriented dialogue agent for the MultiWOZ 2.0 dataset. For the generative pre-trained language model, we use DistilGPT2 (Sanh et al., 2019), a distilled version of GPT-2. Figure 2 shows the architecture of our policy and critic network based on GPT-2. We design the parameterization of the critic network to share the parameters of the Transformer layers of GPT-2, where the parameters of the Transformer layers are only updated during the policy improvement step. For the hyperparame-

|  | Inform | Success |
|---|---|---|
| TRAINING DATA | 92.10 | 80.45 |
| UPDATED DATA (1) | 93.76 | 83.40 |
| UPDATED DATA (2) | 95.35 | 86.04 |
| UPDATED DATA (3) | **95.85** | **87.05** |

Table 1: The performance of training dataset and self-generated dialogues being used for each iteration of policy improvement step.

|  | Inform | Success | BLEU | Combined Score |
|---|---|---|---|---|
| POLICY ITERATION (1) | 87.47 | 74.43 | 17.61 | 98.56 |
| POLICY ITERATION (2) | 88.60 | 76.43 | 17.72 | 100.23 |
| POLICY ITERATION (3) | 89.83 | **77.33** | 17.36 | 100.95 |
| POLICY ITERATION (4) | **90.07** | 76.63 | **17.78** | **101.13** |

Table 2: Experimental results for the policy improvement with GPT-Critic on MultiWOZ 2.0 dataset.

ters of fine-tuning the GPT-2 model, we follow the setting in the public code of UBAR (Yang et al., 2021). We use $N = 5$ for the number of candidate actions $\{a_k\}^N$, and the set of candidate actions are constructed by vanilla softmax sampling from the policy, rather than beam search, to collect diverse actions. For each behavior cloning iteration, all models are fine-tuned with a training dataset from the pre-trained GPT-2 and early stop according to the loss on the validation set.

## 5.2 EVALUATION ON THE MULTIWOZ DATASET

We evaluate our algorithm on the MultiWOZ 2.0 dataset, which is one of the representative task-oriented dialogue benchmarks. The MultiWOZ 2.0 is a large-scale multi-domain Wizard-of-Oz dataset, where a tourist (i.e. user) converses with a clerk (i.e. system) at the information center in a touristic city. It consists of 8438/1000/1000 dialogues for training/validation/testing. For end-to-end evaluation on the MultiWOZ 2.0 dataset, we use the following automatic evaluation metrics: 1) **Inform**: evaluates whether the system provides an appropriate entity, 2) **Success**: evaluates whether the system answers all the requested information, 3) **BLEU**: measures the fluency of the generated response (Papineni et al., 2002). We also report the **Combined Score** as an overall quality measure (Combined = (Inform + Success) × 0.5 + BLEU).

We compare the performance of GPT-Critic with the following algorithms: 1) SFN+RL (Mehri et al., 2019), a seq2seq network that incorporates several pre-trained dialogue modules into a neural dialogue model, 2) DAMD (Zhang et al., 2020), the domain-aware multi-decoder network with multi-action data augmentation method, 3) SimpleTOD (Hosseini-Asl et al., 2020), a GPT-2-based end-to-end dialogue agent that all sub-tasks recast as a single sequence prediction problem, 4) SOLOIST (Peng et al., 2021), a GPT-2-based end-to-end dialogue agent with transfer learning and machine teaching, 5) MinTL (Lin et al., 2020), an efficient dialogue state tracking method with a minimal generation length by predicting the difference between old and new states, 6) UBAR (Yang et al., 2021), a GPT-2-based end-to-end dialogue agent that leverages the entire dialogue session of every dialogue turn. We implement our algorithm into the codebase of UBAR (Yang et al., 2021), and the result of UBAR is reproduced by adapting its code to the same evaluation settings as other papers[1]. Moreover, we also compare the data augmentation method, DATA AUGMENTATION, which is naively fine-tuning the GPT-2 model with additionally generated data by vanilla softmax sampling from the trained policy.

In addition, we also compare with recent offline RL algorithms that are free from the issue of diverging from human language: 1) CRR (Wang et al., 2020), a value-filtered regression method that performs weighted behavior cloning of offline dataset, 2) Decision Transformer (Chen et al., 2021), a Transformer-based architecture that casts the problem of RL as conditional sequence modeling. For a fair comparison, we use the same pre-trained GPT-2 model as a policy network to train the CRR and the Decision Transformer. Moreover, to show that the policy-gradient-based standard RL algorithms suffer from diverging from human language, we also provide examples of responses generated by policy-gradient-based standard RL algorithm in Appendix D.

---

[1]The score reported in the UBAR paper is the result of using the *true* dialogue state for DB search. In order to compare under the same conditions with other algorithms, we record the result of UBAR to use the *predicted* dialogue state for DB search.

| Algorithms | Inform | Success | BLEU | Combined Score |
|---|---|---|---|---|
| SFN+RL* (Mehri et al., 2019) | 73.80 | 53.60 | 16.90 | 83.10 |
| DAMD* (Zhang et al., 2020) | 76.40 | 60.40 | 16.60 | 85.00 |
| SIMPLETOD* (Hosseini-Asl et al., 2020) | 84.40 | 70.10 | 15.01 | 92.26 |
| SOLOIST* (Peng et al., 2021) | 85.50 | 72.90 | 16.54 | 95.74 |
| MINTL* (Lin et al., 2020) | 84.88 | 74.91 | **17.89** | 97.78 |
| UBAR (Yang et al., 2021) | 87.47 | 74.43 | 17.61 | 98.56 |
| DATA AUGMENTATION | 87.27 | 74.23 | 17.33 | 98.08 |
| CRR (Wang et al., 2020) | 87.00 | 74.23 | 16.52 | 97.14 |
| DECISION TRANSFORMER (Chen et al., 2021) | 87.53 | 74.33 | 17.64 | 98.57 |
| GPT-CRITIC | **90.07** | **76.63** | 17.83 | **101.13** |

Table 3: End-to-end response generation results on MultiWOZ 2.0 dataset. The results with * are from original papers. All other results are averaged over three independent runs.

Table 1 and Table 2 show the results of policy iteration. Table 1 shows the performance of training dataset and critic-guided self-generated dialogues being used for each policy improvement step. Table 2 reports the intermediate performance of behavior cloning of training dataset and critic-guided self-generated dialogues in each of the policy iteration. As shown in Table 1 and Table 2, the performance of critic-guided self-generated dialogues is improved gradually; the performance of GPT-Critic is also consistently improved through the behavior cloning of improved dataset.

Table 3 summarizes the overall performance of GPT-Critic and baseline algorithms in *end-to-end response generation* setting, where the *generated* dialogue state and *generated* dialogue act are used for the DB search and response generation. The results show that GPT-Critic achieved the best performance in terms of inform rate, success rate, and combined score. Moreover, the performance of GPT-Critic on the BLEU score matches those of other pre-trained LM-based methods, since GPT-Critic inherits GPT-2's ability to generate human-like responses through the behavior cloning of responses generated by GPT-2. The results show that GPT-Critic improves the task performance of the agent without the issue of diverging from human language. In addition, as can be shown in Table 3, the naive data augmentation is not effective since it will not change the GPT's sampling distribution in principle.

For the results of offline RL baselines, CRR and Decision Transformer show the results that do not diverge from human-language, since their policy is also trained by behavior cloning. However, both algorithms show limited performance because they perform behavior cloning on a fixed dataset. CRR has achieved remarkable success in continuous control tasks by performing weighted behavior cloning of training dataset filtered by critic, but it does not effectively perform in the task-oriented dialogues because of data scarcity. Furthermore, to evaluate the Decision Transformer, we adopt a delayed return where the agent receives the cumulative reward at the end of dialogue, since the agent cannot observe user goal. Therefore, without observing the user goal at test time, Decision Transformer reduces to the behavior cloning of successful dialogues.

## 5.3 EVALUATION ON CONVLAB EVALUATOR

In order to evaluate the performance of dialogue agents in an end-to-end fashion, we conduct simulator-based evaluation on ConvLab (Zhu et al., 2020). ConvLab is an open-source toolkit that enables to build task-oriented dialogue systems and perform an end-to-end evaluation. The simulator-based evaluation is more reliable than dataset-based automatic evaluation because it evaluates the performance while interacting with the user simulator. To interact with dialogue systems, ConvLab provides an agenda-based user simulator (Schatzmann et al., 2007) that consists of a BERT (Devlin et al., 2019) for NLU, a rule-based policy, and a template-based NLG. We compare the performance of GPT-Critic with baseline algorithms interacting with the same user simulator and user goals. We report the results with the following metrics: 1) **Complete**: evaluates whether the system completes the goal, 2) **Success**: evaluates whether all the user requests have been informed and the booked entities satisfy the constraints, 3) **Book**: evaluates how many booked entities satisfy the user constraints, 4) **Inform (Precision / Recall / F1)**: evaluates how many user requests have been informed, 5) **Turn (success / all)**: evaluates the average turn number for successful/all dialogues.

We describe the performance of GPT-Critic and baselines in Table 7. Each algorithm is tested for 1000 runs with randomly sampled user goal. The results show that GPT-Critic achieves the best

| Algorithms | Complete | Success | Book | Inform (P / R / F1) | Turn (succ / all) |
|---|---|---|---|---|---|
| DAMD | 49.40 | 44.40 | 50.78 | 0.55 / 0.60 / 0.55 | 13.72 / 26.82 |
| MINTL | 71.40 | 68.10 | 65.38 | 0.65 / 0.80 / 0.69 | 15.67 / 20.72 |
| UBAR | 79.80 | 74.30 | 80.82 | **0.74** / 0.86 / 0.76 | 14.22 / 18.08 |
| CRR | 78.20 | 72.60 | 82.21 | **0.74** / 0.84 / 0.76 | **13.61 / 17.96** |
| DECISION TRANSFORMER | 81.30 | 75.30 | 83.54 | 0.73 / 0.87 / 0.77 | 14.81 / 18.05 |
| GPT-CRITIC | **84.30** | **77.70** | **85.42** | **0.74 / 0.90 / 0.79** | 16.27 / 19.42 |

Table 4: Experimental results of simulator-based evaluation on ConvLab. All results are averaged over 1000 dialogues.

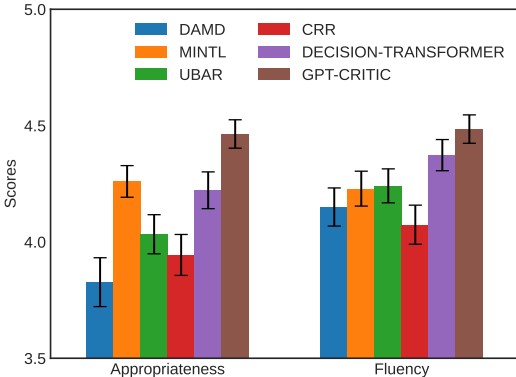

Figure 3: Human evaluation results for appropriateness and fluency. The results represent the mean and standard error for each algorithm.

performance in all metrics related to task accomplishment. However, they also show that GPT-Critic takes longer dialogue turn for the task accomplishment because GPT-Critic is trained by maximizing the success rate without considering the dialogue turn.

## 5.4 HUMAN EVALUATION

We also conduct human evaluation on Amazon Mechanical Turk (AMT) to assess the quality of generated responses of GPT-Critic and baseline algorithms, using the evaluation protocol as in (Yang et al., 2021; Lin et al., 2020; Zhang et al., 2020). Specifically, human workers on AMT were asked to read the context and generated response by interactive simulation via ConvLab, then score the following two evaluation metrics on a Likert scale (1-5): 1) **Appropriateness**: evaluates whether the generated responses are appropriate for the given context, 2) **Fluency**: evaluates whether the generated responses are comprehensible and human-like. We compare the performance of GPT-Critic with same baselines on ConvLab evaluation. Figure 3 summarizes the overall results of human evaluation, where 60 workers evaluate the quality of 30 randomly selected dialogues for each algorithm. The results show that GPT-Critic significantly outperforms baseline algorithms in appropriateness which is related to task accomplishment. Moreover, the result of fluency shows that GPT-Critic does not hurt the agent's capability to generate human-like sentences.

## 6 CONCLUSION

We presented GPT-Critic, an offline RL algorithm for task-oriented dialogue system, which can be adopted for any generative pre-trained language model. GPT-Critic aims to learn an end-to-end task-oriented dialogue agent without the issue of diverging from human language. GPT-Critic starts with fine-tuning the GPT-2 model and learning the critic using the dialogue corpus. Then, GPT-Critic updates the policy through the behavior cloning of the critic-guided self-generated responses, thus it is essentially free from the issue of diverging from human language. In the experiments, we demonstrated that GPT-Critic outperforms the state-of-the-art algorithms in the task-oriented dialogue benchmarks including MultiWOZ 2.0 and ConvLab.

ACKNOWLEDGMENTS

This work was supported by the National Research Foundation (NRF) of Korea (NRF-2019R1A2C1087634, NRF-2021M3I1A1097938) and Institute of Information & communications Technology Planning & Evaluation (IITP) grant funded by the Korea government(MSIT) (No.2019-0-00075, No.2020-0-00940, No.2021-0-02068)

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

## A    POLICY IMPROVEMENT THEOREM

**Theorem 1.** *(Policy Improvement) Given a policy $\pi$ and the number of sampling actions $N \geq 1$, If we update the new policy $\pi_N^{\text{new}}$ by*

$$\forall s, \pi_N^{\text{new}}(\cdot|s) = \arg\max_{\substack{a \in \{a_k\}^N \\ \{a_k\}^N \sim \pi(s)}} Q^\pi(s,a)$$

*then $Q^{\pi_N^{\text{new}}}(s,a) \geq Q^\pi(s,a)\ \forall s,a$ always holds. Furthermore, for any $N, M$ such that $N \geq M \geq 1$, $Q^{\pi_N^{\text{new}}}(s,a) \geq Q^{\pi_M^{\text{new}}}(s,a)\ \forall s,a$ always holds.*

*Proof.* For any $s, a, N \geq M$,

$$Q^{\pi_M^{\text{new}}}(s,a) = \mathbb{E}_P\Big[R(s_t,a_t) + \gamma\mathbb{E}_{a_{t+1}\sim\pi_M^{\text{new}}(s_{t+1})}[Q^{\pi_M^{\text{new}}}(s_{t+1},a_{t+1})]|s_t=s,a_t=a\Big]$$

$$= \mathbb{E}_P\Big[R(s_t,a_t) + \gamma\mathbb{E}_{\{a_i\}^M\sim\pi(s_{t+1})}[\max_{a'\in\{a_i\}^M} Q^\pi(s_{t+1},a')]|s_t=s,a_t=a\Big]$$

$$\leq \mathbb{E}_P\Big[R(s_t,a_t) + \gamma\mathbb{E}_{\{a_i\}^N\sim\pi(s_{t+1})}[\max_{a'\in\{a_i\}^N} Q^\pi(s_{t+1},a')]|s_t=s,a_t=a\Big]$$

$$= \mathbb{E}_P\Big[R(s_t,a_t) + \gamma\mathbb{E}_{a_{t+1}^{\text{new}}\sim\pi_N^{\text{new}}(s_{t+1})}[Q^\pi(s_{t+1},a_{t+1}^{\text{new}})|s_t=s,a_t=a\Big]$$

$$= \mathbb{E}_P\Big[R(s_t,a_t) + \gamma\Big(\mathbb{E}_{a_{t+1}^{\text{new}}\sim\pi_N^{\text{new}}(s_{t+1})}[R(s_{t+1},a_{t+1}^{\text{new}})] + \gamma\mathbb{E}_{a_{t+2}\sim\pi(s_{t+2})}[Q^\pi(s_{t+2},a_{t+2})]\Big)|s_t=s,a_t=a\Big]$$

$$= \mathbb{E}_P\Big[\sum_{\tau=t}^{t+1}\mathbb{E}_{a_{\tau+1}\sim\pi_N^{\text{new}}(s_{\tau+1})}[\gamma^{\tau-t}R(s_\tau,a_\tau)] + \gamma^2\mathbb{E}_{a_{t+2}\sim\pi(s_{t+2})}[Q^\pi(s_{t+2},a_{t+2})]|s_t=s,a_t=a\Big]$$

$$\leq \mathbb{E}_P\Big[\sum_{\tau=t}^{t+1}\mathbb{E}_{a_{\tau+1}\sim\pi^{\text{new}}(s_{\tau+1})}[\gamma^{\tau-t}R(s_\tau,a_\tau)] + \gamma^2\mathbb{E}_{a_{t+2}^{\text{new}}\sim\pi^{\text{new}}(s_{t+2})}[Q^\pi(s_{t+2},a_{t+2}^{\text{new}})|s_t=s,a_t=a\Big]$$

$$= \mathbb{E}_P\Big[\sum_{\tau=t}^{t+2}\mathbb{E}_{a_{\tau+1}\sim\pi^{\text{new}}(s_{\tau+1})}[\gamma^{\tau-t}R(s_\tau,a_\tau)] + \gamma^3\mathbb{E}_{a_{t+3}\sim\pi(s_{t+3})}[Q^\pi(s_{t+3},a_{t+3})|s_t=s,a_t=a\Big]$$

$$\vdots$$

$$\leq \mathbb{E}_P\Big[\sum_{\tau=t}^{\infty}\mathbb{E}_{a_{\tau+1}\sim\pi^{\text{new}}(s_{\tau+1})}[\gamma^{\tau-t}R(s_\tau,a_\tau)]|s_t=s,a_t=a\Big]$$

$$= Q^{\pi_N^{\text{new}}}(s,a)$$

For the case of $N = 1$, note that $\pi_1^{\text{new}}$ is simply reduced to the $\pi$, which concludes the proof:

$$Q^{\pi_N^{\text{new}}}(s,a) \geq Q^{\pi_M^{\text{new}}}(s,a) \geq Q^{\pi_1^{\text{new}}}(s,a) = Q^\pi(s,a) \text{ for all } s,a, N \geq M \geq 1.$$

$\square$

# B    Qualitative Analysis of Self-generated Dialogues

| WOZ20322 | Original Dialogue / Self-generated Dialogue |
|---|---|
| User Utterance | i would like a moderately prices restaurant in the east part of town. |
| | i would like the phone number and postcode, please. |
| Dialogue State | [restaurant] pricerange moderate area east |
| Dialogue Act | [restaurant] [select] food [inform] choice area |
| System Response | we have [value_choice] matching your request in the [value_area] part of town. |
| | do you prefer [value_food] or [value_food]? |
| Dialogue Act * | [restaurant] [inform] food price area phone name postcode |
| System Response * | [value_name] is a [value_price] [value_food] restaurant in the [value_area] part of |
| | town. their phone number is [value_phone] and their postcode is [value_postcode]. |

| SNG01237 | Original Dialogue / Self-generated Dialogue |
|---|---|
| User Utterance | i am looking for the nearest police station |
| Dialogue State | - |
| Dialogue Act | [police] [inform] phone |
| System Response | the nearest police station is in parkside cambridge, the telephone number |
| | is [value_phone]. |
| Dialogue Act * | [police] [inform] name phone address postcode |
| System Response * | the nearest police station is [value_name], located in [value_address] [value_postcode]. |
| | their phone number is [value_phone]. is there anything else i can help you with? |

| PMUL2640 | Original Dialogue / Self-generated Dialogue |
|---|---|
| User Utterance | i heard cityroomz is a good hotel. can you give me some information on this hotel? |
| Dialogue State | [hotel] name cityroomz |
| Dialogue Act | [hotel] [inform] area price |
| System Response | it is located in the [value_area] and is priced in the [value_price] range. |
| Dialogue Act * | [hotel] [inform] area name internet price parking type stars [offerbook] |
| System Response * | [value_name] is a [value_type] in the [value_area]. it is [value_price], has [value_stars] |
| | stars, and offers free wifi and parking. would you like me to book it for you? |

| MUL2024 | Original Dialogue / Self-generated Dialogue |
|---|---|
| User Utterance | i am looking for a hotel called the a and b guest house. can you help me out? |
| Dialogue State | [hotel] name a and b guest house |
| Dialogue Act | - |
| System Response | yes, what would you like to know about it? |
| Dialogue Act * | [hotel] [inform] area address name price |
| System Response * | [value_name] is located in the [value_area] at [value_address]. it is [value_price]. |

Table 5: Examples of the original dialogue and the critic-guided self-generated dialogue on the training dataset of MultiWOZ. The dialogue act and system response with * denote examples of self-generated actions (dialogue act and system response) by GPT-Critic. All system responses represent delexicalized responses.

In this section, we provide a qualitative analysis on the critic-guided self-generated responses in GPT-Critic. We show the critic-guided self-generated dialogue examples in Table 5 that illustrates how GPT-Critic improves the performance through the behavior cloning of self-generated dialogues, compared with unsuccessful dialogues on the training dataset of MultiWOZ. Each example demonstrates the critic-guided self-generated dialogue act and delexicalized system response, compared with the original dialogue. The generated responses contain all the requests of user with abundant information, whereas the original responses of unsuccessful dialogues do not contain all the requested information. GPT-Critic improves the performance through the behavior cloning of these

revised responses. Moreover, Table 5 shows that the generated dialogues do not diverge from human language. Since GPT-Critic updates the policy through behavior cloning of the self-generated human-like responses, GPT-Critic is essentially free from the issue of diverging from human language.

## C  QUALITATIVE EXAMPLES OF STANDARD REINFORCEMENT LEARNING ALGORITHM

In this section, we provide examples of responses generated by standard RL algorithm (REINFORCE) to show that the policy-gradient-based standard RL algorithms suffer from diverging from human language. As can be shown in Table 6, policy-gradient-based RL algorithm generates responses which is diverging from human language.

| WOZ20010 | Original Dialogue / Generated Dialogue |
|---|---|
| User Utterance | i need a restaurant. |
| Dialogue State | - |
| Dialogue Act | [restaurant] [request] area |
| System Response | in what part of town? |
| Dialogue Act * | [restaurant] [request] name |
| System Response * | do you know what you are looking for? i can direct you to something you will need. if you think of it, please let me know. goodbye. i hope everything works. thank you for calling. goodbye. i hope everything works. |
| **PMUL0799** | **Original Dialogue / Generated Dialogue** |
| User Utterance | awesome. i also need a place to eat in the moderate range. i am thinking maybe in the city centre. |
| Dialogue State | [restaurant] pricerange moderate area centre [train] day wednesday arrive 17:00 destination cambridge departure london liverpool street people 1 |
| Dialogue Act | [restaurant] [select] [inform] area food choice price |
| System Response | there are [value_choice] [value_price] restaurant s in the [value_area], 4 serve [value_food], and 1 serves [value_food] food. which would you prefer? |
| Dialogue Act * | - |
| System Response * | alright, have a wonderful day! you have a wonderful day as well. have a great day!!!!! you have a nice day as well!!!!!!!!!!!!!!! |
| **PMUL3700** | **Original Dialogue / Generated Dialogue** |
| User Utterance | i need a hotel and i need free wifi and parking. |
| Dialogue State | [hotel] parking yes internet yes |
| Dialogue Act | [hotel] [inform] choice [request] stars type |
| System Response | i have found [value_choice] place -s that fit your criteria. to further narrow your search, please provide a specific hotel type and the amount of stars of the hotel you would like. |
| Dialogue Act * | hemp bar ask ask ask ask ask ask ask ask ask ask ask ask ask ask cause get call us carizza our service chatosa! we baill us love!WeXT do!Win! |
| System Response * | DrawLineWeTact have us online - we rock! sign us hemp! sign us hemp! us hemp! sign us hemp! sign us hemp! sign us hemp! sign us hemp! sign us hemp! sign us hemp |

Table 6: Examples of the original dialogue and the generated dialogue by policy-gradient-based RL algorithm (REINFORCE) on the training dataset of MultiWOZ. The dialogue act and system response with * denote examples of generated actions by REINFORCE. All system responses represent delexicalized responses.

## D    EVALUATION FOR THE QUALITY OF GENERATED DIALOGUE STATES AND DIALOGUE ACTS

| Algorithms | Joint accuracy | Slot accuracy | Dialogue act f1 |
|:---:|:---:|:---:|:---:|
| UBAR | $47.50 \pm 1.39$ | $95.77 \pm 0.13$ | $50.94 \pm 0.17$ |
| GPT-CRITIC | $47.37 \pm 1.14$ | $95.80 \pm 0.11$ | $\mathbf{51.81 \pm 0.09}$ |

Table 7: Experimental results for the quality of generated dialogue states and dialogue acts on MultiWOZ 2.0 dataset.

We additionally conducted the experiments with UBAR and GPT-Critic to explicitly evaluate the generated dialogue states and dialogue acts (rather than evaluating the final system response). The table below shows the performance for predicted dialogue state(Joint accuracy / Slot accuracy) and predicted dialogue act (Dialogue Act f1), where the mean performance and the standard error are reported. As the table presents, GPT-Critic outperforms UBAR on Dialogue Act f1, which measures the performance of dialogue policy prediction. However, in the case of dialogue state tracking, there is no significant performance gap between GPT-Critic and UBAR since our GPT-Critic revises only the dialogue act and system response (which are considered as an action in GPT-Critic) but not the dialogue state in the dataset.

