# OpenReview forum: "GPT-Critic: Offline Reinforcement Learning for End-to-End Task-Oriented Dialogue Systems"
_ICLR.cc/2022/Conference — ICLR 2022 Poster_

### Official Review · Reviewer_JTos · 2021-10-24

**Correctness:** 3
**Technical Novelty And Significance:** 3
**Empirical Novelty And Significance:** 3
**Recommendation:** 6
**Confidence:** 4

**Main Review:**

The paper proposes a nice way to handle large scale natural language action space for RL. An action in this model is a system action plus response (a sequence of tokens) rather than pre-specified variables in discrete latent models for dialogue (LaRL, etc.), so the proposed model can be easily incorporated into a transformer-based language model (e.g. GPT-2).

My biggest concern is the q-value estimation. Estimating the Q value for an offline dataset is hard due to the problem of overestimation, which is already pointed out by the authors in the paper. However, the authors didn't describe in detail how they mitigate this problem. The only sentence in the paper about this is "However, we avoid this OOD problem by xxx ... revised by generated system response and evaluated reward using offline automatic evaluation". More details are needed, and ablation study about this is needed. Otherwise, I don't know if the proposed method can generalize to other natural language action space applications where the ground-truth reward function is unknown for offline automatic evaluation.

My second concern is the exploration perspective of self-generation. The paper mentioned that {a_k}^N is a set of N response candidates generated from the current policy \pi. What number is N set to? What is the impact of N to the final performance? Moreover, is{a_k}^N sampled by beam search or vanilla sampling? I'd assume that we want to sample {a_k}^N that is diverse enough, rather than similar system responses with only one or two different words. More discussion about the generation is needed.



**Summary Of The Paper:**

The paper works on offline reinforcement learning for natural language action space setting, particularly for task-oriented dialogue management. The paper nicely incorporate the policy network (to sample agent response) and the q network (to evaluate the agent response) into a single GPT-2 network and propose a policy interation algorithm to optimize both the q and policy network.
During policy evaluation, the q network is updated with sampled system actions and responses. During policy improvement, the sampled system actions and responses with maximum q values are used as labels to update the policy network. The model achieves SoTA performance on MultiWOZ dataset.

**Summary Of The Review:**

1. Good method to incorporate policy network and q network into a GPT-2 model to handle large scale natural language action space.
2. Have concern on offline q-value estimation and sampling of response candidates.

---

> ### Author Response · Authors · 2021-11-20
> **Response to Reviewer JTos**
>
> Thank you for your constructive feedback and comments. Please feel free to ask any additional follow-up questions.
>
> 1-1. (mitigating overestimation) Please note that GPT-Critic performs **on-policy evaluation** on the dataset, which can be optimized very stably since every $a_{t+1}$ is always an in-distribution sample of the dataset. After the policy evaluation step, GPT-Critic performs policy improvement by generating an improved dataset based on the learned critic, where we can perform the on-policy evaluation on the new dataset again. Since GPT-Critic **only performs on-policy evaluation** for the policy iteration, we are essentially free from Q-value overestimation. This is in contrast to prior offline RL algorithms, which require evaluation of out-of-distribution actions sampled from the target policy. We revised the main method sections (Section 2.1 and 3.1) of the paper for clarification.
>
> 1-2. (Reward function) Our GPT-Critic assumes a ground-truth reward function for offline automatic evaluation. Please note that task-oriented dialogue domains commonly provide a ground reward function, e.g. MultiWOZ, since it is straightforward to automatically determine task success or failure by looking at the user goal and the dialogue history. Currently, we limit ourselves to task-oriented dialogues. We have changed the title of the paper accordingly.
>
> 2-1. (The number of candidate actions) For our experiments, we used $N=5$ for the number of candidate actions. Theoretically, we can show that the policy updated with larger N has a higher value than the policy updated with smaller N. Then, as we increase N, better performance is guaranteed. We added the discussion regarding the impact of the number of candidate actions to (Section 3.2) in the revised paper.
>
> 2-2. (Sampling method) The candidate actions ${a_k}^N$ are generated by vanilla softmax sampling from the trained policy since we observed that generated candidate actions are diverse enough with vanilla softmax sampling. We added clarification regarding the details of generation to (Section 5.1) in the revised paper.

---

> > ### Public Comment · ~Luyao_Yuan1 · 2022-06-02
> > **Why it is on-policy?**
> >
> > From Equation 2, it is indeed on-policy. However, if we consider how the dataset is generated, it uses argmax Q, instead of actions directly sampled from \pi, which effectively makes this off-policy just like Q-learning.

---

### Official Review · Reviewer_fa2f · 2021-10-24

**Correctness:** 4
**Technical Novelty And Significance:** 1
**Empirical Novelty And Significance:** 2
**Recommendation:** 6
**Confidence:** 5

**Main Review:**

Strengths:
This paper introduces a critic value function on the top of the pretrained GPT-2 task-oriented dialogue agent, to guide the response generation.

Weakness:
1. The novelty may be limited by only adding a critic value function on the top of the existing work.

2. Is it possible to add human evaluation for at least of the datasets? (fixed in the rebuttal)


**Summary Of The Paper:**

To tackle the response generation diverging issue from human language, this paper introduces a critic on the top of the pretrained GPT-2 task-oriented dialogue agent, and demonstrates promising empirical results on two datasets (MultiWoz and ConvLab).

**Summary Of The Review:**

The motivation (tackling the response generation diverging issue from human language) of this work is clear, and the solution is also good, by adding a critic on the top of existing pre-trained GPT-2 dialogue agent, and shows promising empirical results on the automatic evaluation of two datasets. However, the main concern for this work, is is quite incremental with limited novelty, and also lacks human evaluation to verify its effectiveness, may recommend for a workshop paper.

---

> ### Author Response · Authors · 2021-11-20
> **Response to Reviewer fa2f**
>
> Thank you for your constructive feedback and comments. Please feel free to ask any additional follow-up questions.
>
> 1. To show our contribution and novelty more clearly, we revised the main method section (Section 3) of the paper for clarification.
> - Our motivation and novelty are as follows:
> The weighted behavior cloning (BC) is one of the representative offline RL algorithms, which is free from the issue of diverging from human language. The weighted BC amounts to filtering out bad actions and imitating good actions.  In the context of task-oriented dialogues, that would be similar to simply dropping the unsuccessful dialogues from the corpus. Dropping a whole dialogue from training would be wasteful, since they may still contain some task-specific information that is useful to properly respond to user requests in the intermediate steps. Our GPT-Critic aims to **revise unsuccessful dialogues into successful ones**, rather than removing them as done in weighted BC. Compared to traditional actor-critic methods, the policy improvement is carried out **implicitly** by generating a better dataset to be imitated, rather than **explicitly** taking the policy gradient.
> - Our main contributions are as follows: 1) GPT-Critic is essentially free from the issue of diverging from human language, unlike policy-gradient-based actor-critic methods, because GPT-Critic is basically an iterative behavior cloning of the human-like dialogues generated from GPT-2, 2) GPT-Critic performs policy improvement by **generating an improved dataset** based on the critic, where we can perform the on-policy evaluation on the revised dataset. As a consequence, GPT-Critic can enjoy the stable **multi-step** policy iteration through alternation between on-policy evaluation and policy improvement via revising the dataset.
>
> 2. We added additional experimental results for the human evaluation. We conducted a human evaluation to evaluate the quality of the generated response by interactive simulation via ConvLab. We use the following evaluation metrics: 1) Appropriateness: evaluates whether the generated responses are appropriate for the given context, 2) Fluency: evaluates whether the generated responses are comprehensible. We observed that GPT-Critic generates more appropriate responses (appropriateness) than other algorithms without hurting the ability to generate human-like responses (fluency). This new result and details of experimental settings are added to (Section 5.4, Figure 3) in the revised paper.

---

### Official Review · Reviewer_u1n3 · 2021-11-03

**Correctness:** 3
**Technical Novelty And Significance:** 2
**Empirical Novelty And Significance:** 3
**Recommendation:** 6
**Confidence:** 3

**Main Review:**

The proposed approach is very simple and should be easy to implement. The experimental results seem promising. However, I do have a couple of concerns.

Firstly, I think that some important details are missing in the paper. For example, what is the action space of the agent? Do you treat the conjunction of a dialogue act and a system response as a single action? If so, how exactly are the candidate actions generated? Is some kind of beam search employed? I think some actual examples of actions (and rewards) would help the reader understand the proposed method more clearly.

The paper does not really describe how the rewards are computed, either. In particular, I am wondering how the reward for the newly selected action is computed. Is it given to the agent by an external program?  Then, it seems to me that the whole training procedure is more like (a somewhat restricted version of) actor-critic-based reinforcement learning than offline reinforcement learning (in which the agent cannot interact with the environment). If that is the case, what is the novelty of the proposed method?

The authors claim in Table 3 that their proposed approach gives much better results than UBAR, but the original paper of UBAR (Yang et al., 2021) reports much better results (e.g., Inform score of 95.4). Why is there such a big difference?

Algorithm 1 states that the policy is updated by behavior cloning until "convergence". I am wondering if it causes any overfitting problem. Is overfitting the reason why the whole training process is stopped at the fourth iteration (Table 2)?

Minor comments:

p. 1: outperforms the state-of-the-art -> outperforms the state of the art;
p. 1: not trained to for -> not trained for?
p. 2: fine-turning the GPT-2 -> fine-turning GPT-2? fine-turning the GPT-2 model?
p. 2: generates strategically -> generates a strategically;
p. 2: Pr(O_{t+1} ... ) -> should not be italic?
p. 3: by training action-value -> by training the action-value;
p. 3: of critic network -> of the critic network;
p. 4: for $i$-th -> for the $i$-th;
p. 4: in task-oriented -> in the task-oriented?
p. 4: generated system response -> generated system responses?
p. 4: from (Zhao et al., 2019) -> from Zhao et al. (2019);
p. 4: prohibitory -> prohibitively?
p. 4: over response candidates -> over the response candidates?
p. 4: updated policy by above -> the updated policy by the above?
p. 5: on MultiWOZ domain -> on the MultiWOZ domain;
p. 5: ConvLab framework -> the ConvLab framework;
p. 5: HuggingFace Transforms library -> the HuggingFace …;
p. 7: prices -> priced?
p. 8: user goal -> the user goal?
p. 8: with following -> with the following;
p. 8: the all -> all the;
p. 8: whereas original -> whereas the original?
p. 9: straightforward -> straightforwardly?
p. 8: large scale -> large-scale?


**Summary Of The Paper:**

This paper presents a reinforcement learning-based approach to building a task-oriented dialogue agent. Given a dialogue dataset annotated with rewards, the state-action value function is first trained by minimizing temporal differences. Then, a new training dataset is created by using the best actions selected among the candidates generated by the current policy (i.e., the language model). The policy is then updated by behavior cloning using the created dataset, and the whole process is repeated. The authors have conducted experiments using MultiWOZ and ConvLab and shown that this iterative process improves the performance of the agent.

**Summary Of The Review:**

The proposed algorithm is simple and seems effective. However, some important details are missing and the novelty is not very clear.

---

> ### Author Response · Authors · 2021-11-20
> **Response to Reviewer u1n3**
>
> Thank you for your constructive feedback and comments. Please feel free to ask any additional follow-up questions.
> 1. An action of the agent is a sequence of tokens that represents system action (dialogue act) and system response ([1,2]). We treat the conjunction of a dialogue act and a system response as a single action, which is represented by $a_t$ in the paper. Table 5 shows actual examples of actions generated by GPT-Critic, and the candidate actions are generated by vanilla softmax sampling from the trained policy. We added the detailed explanation to (Section 2.1, Section 5.1, and Table 5) in the revised paper.
>
> [1] Tiancheng et al, Rethinking Action Spaces for Reinforcement Learning in End-to-end Dialog Agents with Latent Variable Models, NAACL 2019
>
> [2] Shikib et al, Structured Fusion Networks for Dialog, Sigdial 2019
>
> 2. The reward is given by the external program provided as a part of the MultiWOZ dataset, which checks whether the user goal is satisfied by examining the dialogue history. In task-oriented dialogues, this is a common approach for automatic evaluations of dialogue systems, where the agent generates dialogue state, dialogue act, and system response on every system turn with fixed user utterances. We adopt this evaluation scheme to generate the improved dataset, then finally perform the policy improvement by an improved dataset.
> We argue that our method is an offline RL method because the policy is learned only through the previously collected dataset without any additional interaction with the user (=environment).
>
> To show our novelty and contribution more clearly, we revised the main method section (Section 3) of the paper for clarification.
> - Our motivation and novelty are as follows:
> The weighted behavior cloning (BC) is one of the representative offline RL algorithms, which is free from the issue of diverging from human language. The weighted BC amounts to filtering out bad actions and imitating good actions.  In the context of task-oriented dialogues, that would be similar to simply dropping the unsuccessful dialogues from the corpus. Dropping a whole dialogue from training would be wasteful, since they may still contain some task-specific information that is useful to properly respond to user requests in the intermediate steps. Our GPT-Critic aims to **revise unsuccessful dialogues into successful ones**, rather than removing them as done in weighted BC. Compared to traditional actor-critic methods, the policy improvement is carried out **implicitly** by generating a better dataset to be imitated, rather than **explicitly** taking the policy gradient.
>
> - Our main contributions are as follows: 1) GPT-Critic is essentially free from the issue of diverging from human language, unlike policy-gradient-based actor-critic methods. 2) GPT-Critic performs policy improvement by **generating an improved dataset** based on the learned critic, where we can perform the on-policy evaluation on the new dataset again. As a consequence, GPT-Critic can enjoy the stable **multi-step** policy iteration through alternation between on-policy evaluation and policy improvement via revising the dataset.
>
> 3. The score reported in the UBAR paper is the result of using the **true** dialogue state for DB search. The author's public code shows that the **true** dialogue state is used instead of the **predicted** dialogue state for the DB search [3]. For fair comparison, we fixed the UBAR code to use the **predicted** dialogue state for DB search. We added the explanation to (Section 5.2 and footnote) in the revised paper.
>
> [3] https://github.com/TonyNemo/UBAR-MultiWOZ/issues/3
>
> 4. For each behavior cloning iteration, all models are fine-tuned with a revised dataset from the initial pre-trained GPT-2 and adopted early stopping according to the loss on the validation set (i.e. not “until convergence”). Therefore, there is no overfitting issue, and we observed that the performance converges after the 4th iteration without significant improvement. We added clarification regarding the learning detail of behavior cloning to (Algorithm 1 and Section 5.1) in the revised paper.
> We revised the typos in the paper based on your comments.

---

> > ### Comment · Reviewer_u1n3 · 2021-11-29
> > **The paper is much clearer now**
> >
> > Thank you for the response and revision of the manuscript. Most of my concerns have been resolved by the revision, so I have raised my score to "marginally above the acceptance threshold".

---

### Official Review · Reviewer_ujyM · 2021-11-04

**Correctness:** 3
**Technical Novelty And Significance:** 3
**Empirical Novelty And Significance:** 3
**Recommendation:** 6
**Confidence:** 3

**Main Review:**

This paper focuses on improving the dialogue policy together with the responses by utilizing a pre-trained language model and offline RL.
The claimed contributions include:
1) The proposed method is free from the common issue of diverging from human language, because it learns from the sentences sampled from the pre-trained LM.
2) The proposed method outperforms other SOTA models in offline and interactive online settings, MultiWOZ and ConvLab respectively.

The proposed method is reasonable and moderately novel. The experimental results are promising for both settings.
However, there are unclear parts to be addressed or clarified.

- Because the policy learning procedure utilizes the additionally generated dialogue acts and corresponding responses, it is easy to think that naively fine-tuning the GPT-2 model on the additional generated data may also improve the dialogue model performance in terms of its policy and responses (similar to a data augmentation method). Did the authors try this as another compared baseline? This method should be included in the experiments in order to justify the proposed RL approach is necessary.

- The paper mentioned that the standard RL methods easily fail and generate responses diverging from human language, even when fine-tuning a pre-trained LM. Hence, it will be better to additionally include the results of other standard RL algorithms for better justifying this claim. (Current experiments only include the results of models that are free from this issue.)

- In the experiments in MultiWOZ, this paper only evaluates the response generation results. However, evaluating dialogue policy is also important to justify the learned policy is suitable. It is unclear why the authors only show the response generation results.

- The experiments contain two setups, one is offline response evaluation via MultiWOZ, and another is interactive simulation via ConvLab. Both settings show the better performance of the proposed method. The paper can be better if adding the real-user interactions, because the performance may be different between the simulation environment and the real-user interactions reported by prior results (DSTC in ConvLab). Conducting real-human interactions can better justify the effectiveness of the proposed RL method in practical scenarios.

In sum, the proposed method is relatively novel and the idea is reasonable. The performance seems promising in both settings.
However, the paper does not include detailed descriptions about the proposed method, making readers not easy to understand.
Also, some additional experiments need to be added in order to better justify its claims.

**Summary Of The Paper:**

This paper proposes an offline RL method applied to an end-to-end task-oriented dialogue model, where the proposed GPT-Critic is built on GPT-2 and fine-tuned on the self-generated sentences for policy updating.
The paper claims that it is free from the issue of diverging from human language (a common issue in standard RL), because it learns from the sentences directly sampled from the pre-trained language model.
The conducted experiments show that the proposed model achieves better performance compared to other task-oriented end-to-end dialogue models in both offline and online settings (MultiWOZ and ConvLab respectively).

**Summary Of The Review:**

The idea is reasonable and moderately novel. The conducted experiments demonstrate the better performance of the proposed model in two different settings.
The paper misses some details when describing the proposed method, making readers difficult to fully understand its idea.
Although the experiments already include a lot of baselines, there are still some results to be included for fair comparison (simple data augmentation method).

---

> ### Author Response · Authors · 2021-11-20
> **Response to Reviewer ujyM**
>
> Thank you for your constructive feedback and comments. Please feel free to ask any additional follow-up questions.
>
> 1. Thank you for your suggestion for the additional baseline using a simple data augmentation method. We added an additional result of the suggested baseline which is naively fine-tuning the GPT-2 model with additionally generated data by vanilla softmax sampling from the trained policy. As can be seen from the revised paper (Table 3), this naive data augmentation is not effective since it will not change the GPT's sampling distribution in principle. Our GPT-Critic still significantly outperforms this baseline, which demonstrates the effectiveness of our method that leaves only promising actions via the critic.
>
> 2. We also conducted additional experiments for the standard RL method (UBAR+REINFORCE), which applies REINFORCE to the UBAR. We observed that UBAR+REINFORCE suffered from diverging from human language (e.g. repeating the same words or generating incomprehensible sentences), even though it uses a pre-trained language model. We added the results and generated examples to (Appendix C) in the revised paper.
>
> 3. Table 3 shows the end-to-end response generation results on MultiWOZ, where each model outputs the system response using the **generated** dialogue state and the **generated** dialogue-act, not using the ground-truth dialogue state nor dialogue-act. Therefore, the high score for {Inform, Success, BLEU} in Table 3 implies that all of the dialogue state tracking, the dialogue policy, and the response generation modules should jointly work well. We added clarification regarding the evaluation setting of end-to-end response generation to (Section 5.2) in the revised paper.
>
> 4. In order to complement the limitation of automatic evaluation, we added additional experimental results for the human evaluation. We conducted a human evaluation to evaluate the quality of the generated response by interactive simulation via ConvLab. We use the following evaluation metrics: 1) Appropriateness: evaluates whether the generated responses are appropriate for the given context, 2) Fluency: evaluates whether the generated responses are comprehensible. We observed that GPT-Critic generates more appropriate responses (appropriateness) than other algorithms without hurting the ability to generate human-like responses (fluency). This new result and details of experimental settings are added to (Section 5.4, Figure 3) in the revised paper.

---

> > ### Comment · Reviewer_ujyM · 2021-11-28
> > **Evaluation of the generated dialogue states / dialogue acts**
> >
> > Thanks the authors to address some of my concerns by revising the submission.
> >
> > - It is great to see the additional baselines and the proposed method still shows the improved performance.
> > - Although the generated responses implicitly reflect the quality of dialogue states, evaluating the generated dialogue states/acts is still needed. The reason is that the responses are affected by the NLG so that evaluating the dialogue states may be more objective and convincing. Better performance of both dialogue states and responses can better demonstrate the effectiveness of the proposed method.
> > - Adding human evaluation is great and makes the paper more convincing.
> >
> > In sum, the proposed method is reasonable and moderately novel.
> > The conducted experiments also suitably demonstrate the improved performance compared to other work on benchmark data.
> > In my opinion, the paper is above the threshold, but if it failed to be accepted, I will encourage the authors to resubmit to another conference.

---

> > > ### Author Response · Authors · 2021-11-30
> > > **Response to Reviewer ujyM**
> > >
> > > Thank you for suggesting the additional evaluation for the quality of generated dialogue states and dialogue acts. We additionally conducted the experiments with UBAR and GPT-Critic to **explicitly** evaluate the generated dialogue states and dialogue acts (rather than evaluating the final system response). The table below shows the performance for predicted dialogue state(Joint accuracy / Slot accuracy) and predicted dialogue act(Dialogue Act f1), where the mean performance and the standard error are reported. As the table presents, GPT-Critic outperforms UBAR on Dialogue Act f1, which measures the performance of dialogue policy prediction. However, in the case of dialogue state tracking, there is no significant performance gap between GPT-Critic and UBAR since our GPT-Critic **revises only the dialogue act and system response** (which are considered as an action in GPT-Critic) but not the dialogue state in the dataset. We will add this result and analysis to the final version of the paper.
> > >
> > > ||Joint accuracy|Slot accuracy|**Dialogue Act f1**|
> > > |:----------:|:-------------:|:------:|:------:|
> > > |UBAR|47.50 $\pm$ 1.39|95.77 $\pm$ 0.13|50.94 $\pm$ 0.17|
> > > |GPT-Critic|47.37 $\pm$ 1.14|95.80 $\pm$ 0.11|**51.81 $\pm$ 0.09**|

---

### Author Response · Authors · 2021-11-20
**General Response**

We thank all the reviewers for their constructive feedback and comments. Below we restate and highlight the main motivation and contribution of our work. If you have any additional questions or concerns to our response, we are happy to provide additional responses during the rebuttal period.

**[Motivation]**

The weighted behavior cloning (BC) is one of the representative offline RL algorithms, which is free from the issue of diverging from human language. The weighted BC amounts to filtering out bad actions and imitating good actions.  In the context of task-oriented dialogues, that would be similar to simply dropping the unsuccessful dialogues from the corpus. Dropping a whole dialogue from training would be wasteful, since they may still contain some task-specific information that is useful to properly respond to user requests in the intermediate steps. Our GPT-Critic aims to **revise unsuccessful dialogues into successful ones**, rather than removing them as done in weighted BC. Compared to traditional actor-critic methods, the policy improvement is carried out **implicitly** by generating a better dataset to be imitated, rather than **explicitly** taking the policy gradient.

**[Contributions]**

1) GPT-Critic is essentially free from the issue of diverging from human language, unlike policy-gradient-based actor-critic methods, because GPT-Critic is basically an iterative behavior cloning of the human-like dialogues generated from GPT-2
2) GPT-Critic performs policy improvement by **generating an improved dataset** based on the critic, where we can perform the on-policy evaluation on the revised dataset. As a consequence, GPT-Critic can enjoy the stable **multi-step** policy iteration through alternation between on-policy evaluation and policy improvement via revising the dataset.

The detailed improvements and modifications in the revision are summarized as follows:

**[Experiments]**

Additional results of human evaluation to address the suggestions provided by Reviewer 1 and 3 (Section 5.4, Figure 3).
Additional results and analysis for the additional baseline using a simple data augmentation method to address the suggestions provided by Reviewer 1 (Section 5.2, Table 3).
Additional examples to show that the standard RL method suffers from diverging from human language to address the suggestions provided by Reviewer 1 (Appendix C).

**[Writing]**

To limit ourselves to task-oriented dialogues, we have changed the title of the paper to ‘Offline Reinforcement Learning for End-to-End Task-Oriented Dialogue Systems’.
We moved the related work section before the experimental section.
To show our contribution and novelty more clearly, we revised the background, main method, and related work section (Section 2, 3).
Clarification of experimental settings (end-to-end response generation settings, the number of candidate actions, action spaces, sampling method, details of baseline) to address the suggestions provided by Reviewer 1, 2 and 4. (Section 2.1, 5.1, Table 3, Table 5)

---

### Public Comment · ~Yihao_Feng1 · 2022-05-27
**Public Code**

Hi,

Really nice work. I wonder if it is possible to release the code of the amazing work?

---

### Decision · Program_Chairs · 2022-01-20

**Decision:**

Accept (Poster)

**Comment:**

The reviewers are all weakly positive. The author response clarified important aspects of the paper. The new human evaluation was critical. However, the human evaluation result presentation is flawed: presenting Likert scores as means does not reflect them well. The authors should use something similar to a Gantt chart to fully reflect the distribution across Likert categories. Another detail in the human evaluation that are troubling: it does not reflect interaction with the system, but judgements through observation. Therefore, the human evaluation does not reflect the ability of the learned dialogue system to interact with users. Overall, the paper makes a nice, original contribution, but despite author improvement there are evaluation flaws (even if they are common in papers using these benchmarks).